# Spontaneous Primary Succession and Vascular Plant Recovery in the Iberian Gypsum Quarries: Insights for Ecological Restoration in an EU Priority Habitat

**DOI:** 10.3390/plants12051162

**Published:** 2023-03-03

**Authors:** Juan Francisco Mota, Fabián Martínez-Hernández, Esteban Salmerón-Sánchez, Antonio Jesús Mendoza-Fernández, Francisco Javier Pérez-García, M. Encarna Merlo

**Affiliations:** 1Department of Biology and Geology, University of Almería, 04120 Almería, Spain; 2Department of Botany, University of Granada, 18071 Granada, Spain

**Keywords:** gypsophil, gypsum mining, passive restoration, permanent plots, Species-Area Relationships (SAR), successional chronosequence

## Abstract

Gypsum covers a vast area of the Iberian Peninsula, making Spain a leader in its production. Gypsum is a fundamental raw material for modern societies. However, gypsum quarries have an obvious impact on the landscape and biodiversity. Gypsum outcrops host a high percentage of endemic plants and unique vegetation, considered a priority by the EU. Restoring gypsum areas after mining is a key strategy to prevent biodiversity loss. For the implementation of restoration approaches, understanding vegetation’s successional processes can be of invaluable help. To fully document the spontaneous succession in gypsum quarries and to evaluate its interest for restoration, 10 permanent plots of 20 × 50 m were proposed, with nested subplots, in which vegetation change was recorded for 13 years in Almeria (Spain). Through Species-Area Relationships (SARs), these plots’ floristic changes were monitored and compared to others in which an active restoration was carried out, as well as others with natural vegetation. Furthermore, the successional pattern found was compared to those recorded in 28 quarries distributed throughout the Spanish territory. The results show that an ecological pattern of spontaneous primary auto-succession is widely recurring in Iberian gypsum quarries, which is capable of regenerating the pre-existing natural vegetation.

## 1. Introduction

About 1% of the earth’s surface is directly influenced by mining activities [1]. These have devastated large areas of land and reduced their biodiversity. Given the evidence that in most parts of the world, the impact of human beings on biodiversity, ecosystems and ecological processes that sustain them is growing exponentially, it is obvious that the restoration of ecosystems, landscapes and sites affected by mining is necessary. Goal 15 for sustainable development of the United Nations and article 8 of the Convention on Biological Diversity insist on this same idea. Unsurprisingly, the restoration of biodiversity and ecosystem services in post-mined and post-industrial sites has received careful attention among scientists, to the extent that it represents an important part of contemporary restoration ecology [2].

Succession, the sequential and temporal replacement of species after a disturbance, is a key ecological process upon which to base ecological restoration. Although this has been a central theme throughout the 150-year-long history of ecology [3,4,5,6], many restoration professionals are unaware of the use succession may have in planning and goal setting [7]. The lessons that emerge from the study of ecological succession have much to offer for the resolution of contemporary environmental problems [8,9,10,11,12]. According to Prach [13], restoration practices clearly benefit from the results of successional studies. Spontaneous successional processes are an important aspect of ecological restoration and must be taken into account in virtually any restoration programme, to the point that, in some cases, they can be completely relied upon [14]. To identify the role of spontaneous succession in particular restoration programs, it is necessary to: (1) document processes of this type in detail that provide information on their general trends, limits and possibilities; (2) carry out comparative studies and experimental trials that allow for spontaneous succession to be assessed against other alternatives; (3) perform medium- and long-term monitoring of all the alternatives that may be considered, either through permanent monitoring plots or, due to the urgency of the action, using chronosequences or time series [15].

The application of these approaches to daily practice is urgent because the environmental administration needs clear and immediate guidelines [16]. A beneficial application of succession studies is to use them for inspiration in ecological restoration [17], which should be interpreted as the intentional manipulation of succession [18,19]. From the authors’ point of view, restoration is a practical implementation of succession concepts involving operational and field management control [7]. Furthermore, scientific approaches to restoration can also clarify successional processes and improve successional predictability, leading to reciprocal benefits [16].

The successional patterns found in gypsum quarries clearly fit what is called primary succession [20,21], which is triggered when a disturbance is so dramatic that little or no biological legacy remains in the affected terrain [22]. In this case, the disturbance essentially creates a new site for colonization that does not preserve pre-existing conditions [23]. In this type of succession, abiotic factors are especially important, including soil stability and fertility, as well as the dispersal of propagules to the site. The opposite occurs in the processes of secondary succession, which are triggered when there is still a substantial biological legacy; in their temporal development, biotic factors usually stand out, such as the initial presence of seeds and other propagules, biological interactions and the action of microbial communities in the soil [23].

Spontaneous succession in gypsum has other characteristics that make it unique, and has also been considered an example of self-succession or direct succession [24]. However, as Matthews et al. [25] suggested, there are very few cases in which the term autosuccession has been used in the context of primary succession. According to the cited authors, autosuccession was first described by Muller [26]. Building on this idea, Matthews [27,28,29] and Robbins and Matthews [30] explicitly referred to this concept when explaining the differences between the primary successional pathways in the glacial promontories of southern Norway. They concluded, in accordance with Muller [31], that these were autosuccesional processes, although of two different types. Thus, selective autosuccession took place in the middle alpine belt while non-selective autosuccession characterized the more severe environmental conditions of the upper alpine belt. In a very similar sense, Svoboda and Henry [32] recognized two related types of primary succession: the so-called “directional succession without replacement” and the “non-directional succession without replacement”, which they considered to be characteristic of polar semi-deserts and polar deserts, respectively. In his review of successional models in the context of glacial forelands, Matthews [28] noted that Svoboda’s and Henry’s concepts were essentially equivalent to selective and non-selective autosuccession, respectively, and linked all four to the severity of the physical environment in a geo-ecological model of primary succession.

The terms non-selective autosuccession and non-directional succession without replacement seem to coincide with Shreve’s observations [33] who pointed out the “abeyance of successional phenomena” in desert areas, in such a way that after a disturbance the first stage of succession restores the original vegetation directly. This is how Whittaker and Levin [34] interpreted it, to then propose the concept of “direct succession” to describe this ecological process. It is no coincidence that Muller and Shreve presented this idea based on observations of the vegetation in desert areas, although it was later extended to other territories with extreme climates such as the tundra [31]. For Svoboda and Henry [32], only a small group of species being appropriately adapted to colonize these environments and the strong control exerted by abiotic processes are to be found among the key aspects to this pattern. Additionally, the fact that gypsum is a complex and severe substrate for plants [35] suggests that, as in the cases already mentioned, direct primary spontaneous succession may be a very important mechanism when it comes to proposing passive.

Natural regeneration, interpreted in many cases as passive or unassisted restoration (see below), is a recovery process that occurs without active human intervention. In many cases, this requires the removal of persistent disturbances, such as fire or grazing [36]. Compared to active restoration, passive restoration is typically considered an inexpensive or even free alternative [37], and has the potential to achieve similar gains in biodiversity and ecosystem services with less human intervention [38,39]. It also requires little technical experience. Globally, passive restoration plays a much larger role in habitat recovery than active restoration [40,41], and is expected to be a key mechanism for the persistence of biodiversity in the future [42]. Humans actively intervening in an effort to accelerate and influence the successional trajectory of recovery (active restoration) are to be found at the other extreme of restoration approaches. Both perspectives are subject to intense debate [38,43].

The study hereby presented provides detailed information on spontaneous primary autosuccession, after monitoring 10 permanent post-mining plots for 13 years. For this, SARs (species-area relationships) curves and diversity indices were used. The communities generated through this successional process were compared to those of plots associated with active restoration and with the predominant natural or reference vegetation in the area. In this case, the reference ecosystem [44] is represented by the unaltered gypsum scrubland, listed as priority habitat number 1520* (Iberian gypsum steppes, *Gypsophiletalia*) by the EU [45].

Therefore, the objectives of this study were to:Depict the spontaneous successional pattern that takes place in gypsum quarries in southeastern Spain once their exploitation has ceased, and verify that the successional stages experienced a trend towards greater richness and floristic diversity throughout a chronosequence of 13 years. Therefore, the successional SAR curves were expected to converge towards those of the undisturbed gypsum scrublands.In the same way, it was hypothesized that the restored plots (active restoration) would achieve levels of richness, floristic composition, species abundance and diversity similar to those of the unaltered reference scrubland in a few years and it was assumed that these levels would be higher than those of the successional plots even for the final year of monitoring, 2021. According to this hypothesis, the SAR curves of these plots should be closer to those of the reference scrub than the successional ones.Verify whether the same pattern of spontaneous succession as described above is widespread in gypsum quarries throughout the Iberian Peninsula.Draw, from the previous analyses, the implications that both spontaneous succession and active restoration have regarding restoration strategies in an EU priority habitat located in a hotspot of global diversity, especially in that which concerns the gypsicolous vegetation.

## 2. Results

### 2.1. Permanent Successional Plots

#### 2.1.1. Floristic Changes along the Plots in the Chronosequence (Appendix A)

Out of the 183 species that were recorded, only 2, *Gypsophila struthium* subsp. *struthium* and *Sedum gypsicola*, were present in all the floristic inventories. In addition to these, four other species were recorded in more than 90% of the samples: *Bromus rubens*, *Diplotaxis harra* subsp. *lagascana*, *Sedum sediforme* and *Sonchus tenerrimus*. Of the fourteen species considered gypsophiles in the area, only three were not found in any of the plots during the thirteen years of monitoring: *Narcissus tortifolius*, *Rosmarinus eriocalix* and *Campanula fastigiata*. The first two are considered threatened. On the contrary, *Teucrium turredanum* and *Helianthemum alypoides*, the two local endemisms additionally catalogued as VU, were recorded in several of the successional plots. However, the frequency and abundance of these was much lower in these plots than in the undisturbed scrubland. The same may be said for almost all other gypsophytes, especially in the case of *Chaenorrhinum grandiflorum* subsp. *grandiflorum*, *Coris hispanica*, *Helianthemum squamatum*, *Ononis tridentata* and *Santolina viscosa*. Other gypsophytes, such as *Launaea pumila* or the aforementioned *Rosmarinus eriocalix*, although frequent in other gypsum outcrops in the area, are extremely rare in the gypsum area of Sorbas, which might explain their absence from the successional plots. In the case of most perennial gypsophytes, either they have tended to increase their frequency very slightly over the 13 years of monitoring, such as *Santolina viscosa* and *Helianthemum alypoides*, or, like *Helianthemum squamatum* or *Teucrium turredanum*, they have suffered slight abundance fluctuations.

Other very frequent species in the gypsum scrubs of the area such as *Helianthemum syriacum* and *Launaea fragilis*, considered as subgypsophile or gypsocline species, registered a clear increase in their presence throughout the temporal monitoring. On the other hand, *Sedum sediforme* and *Stipa tenacissima*, omnipresent in the gypsicolous scrubland, offered divergent patterns. While *S. sediforme* was already a frequent species in 2009 and remained so throughout the monitoring period, *S. tenacissima* (also *Asparagus horridus*) was only recorded in one or two plots during the last two years. Other species, such as *Anthyllis terniflora* and *Dactylis glomerata*, also abundant in the undisturbed gypsum scrubland, were present from the beginning of the monitoring in half of the plots and their presence continued to increase with time. *Diplotaxis harra* subsp. *lagascana* was ubiquitous from the beginning of the monitoring. As for ruderal species, it is worth highlighting the cases of *Artemisia barrelieri* and *Dittrichia viscosa*. The former is dominant in the oldfields, although it has also been present in the gypsum scrub itself. On the contrary, *D. viscosa* is a ruderal-viary plant with remarkable invasive potential and great dispersal capacity that, however, is not part of the gypsum scrubs. Bearing in mind that both species are ruderal in nature and belong to the family Compositae, the radically different behaviour they presented is striking. While *A. barrelieri* was a late-colonizing species with almost anecdotal presence, *D. viscosa* was present from the beginning of the successional process. This latter pattern was also repeated in the grasses *Piptatherum miliaceum* and *Stipa parviflora*, with which *D. viscosa* is usually associated and can grow into dense vegetation on roadsides and disturbed urban areas [46]. Regarding the bulbous *Gladiolus communis*, *Drimia maritima* or *Moraea sisyrinchium*, almost always present in clearings in the scrubland, their ability to colonize the quarry squares was almost nil. Therophytes, however, showed very different patterns from the bulbous species. Many of them, such as *Brachypodium distachyon*, *Bromus rubens*, *Desmazeria rigida*, *Leontodon longirostris*, *Reichardia tingitana*, *Stipa capensis* or *Vulpia ciliata*, were frequent in plots of spontaneous succession from the beginning of the monitoring, although this did not occur in all cases (e.g., *Asterolinum linum-stellatum*, *Helianthemum salicifolium*, *Linum strictum*, *Plantago ovata* or *Lomelosia stellata*). Other more ruderal therophytes (e.g., *Aegilops geniculata*, *Anacyclus clavatus*, *Hordeum leporinum* or *Lactuca viminea*) were almost always absent in successional plots.

#### 2.1.2. Diversity and SAR Curves in Successional Plots

The progressive increase in the number of species in the plots where the chronosequence of spontaneous succession was recorded was evident throughout the 13 years monitored. The mean number of species increased by 10 units over this period, from 18.8 to 28.6, and there were significant differences between these parameters for the years 2009 and 2021. Additionally, all the SARs lines obtained for the 20 × 50 m plots (130 for the diachronic study of spontaneous succession) showed a strong correlation between the number of taxa and the sampled surface (in all cases R^2^ > 0.9) and they fit the power model well. Figure 1 shows the SAR curves for the years 2009, 2015 and 2021, i.e., the initial year of monitoring, the mid-term and the last. The spatial arrangement of these curves, one on top of the other without there being an intersection between them, very clearly reflects the floristic enrichment of the plots over time. Furthermore, a strong Spearman correlation coefficient (r_s_ > 0.85) was found between the *c* values for each of the 13 SAR curves and the time elapsed since the start of monitoring. This increase was also reflected in coefficient *c* of the regression line equation, which went from 1.31 to 2.21 (Figure 1). For its part, the *z* coefficient remained almost constant, although with a very slight decreasing tendency. Regarding the different indexes of richness and diversity calculated, all of them increased progressively throughout the successional process (Figure 2). This trend was even more striking when only perennial species were considered.

The PERMANOVA analysis also reflected differences for the floristic composition of the successional plots. These differences occurred between three to seven years after the start of monitoring, with a mean value of approximately four years (Table 1). This same analysis, but only considering the perennial species, yielded very similar values (3.932, *p* < 0.0001) and placed the significant differences after five years on average. Using as a variable the order number of each permanent successional plot (1–10), this same analysis showed even more marked differences (Table 2), both considering all the species and only the perennial ones (12.97), and it was significant in all pairwise comparisons (*p* < 0.0001).

### 2.2. Restored and Reference Scrubland Plots: Insights for Ecological Restoration

Although it is evident that the spontaneous succession progressed along the temporal sequence towards greater richness and diversity of species, the truth is that for all the indices studied, the values of the successional plots were significantly lower than those of the active restoration and the undisturbed scrub, even in the last year of monitoring. Comparison of the SAR curves for the three plot types leaves no doubt. The value of *c* is much higher for the reference scrub than for the other two environments studied. However, the same does not occur when this parameter between active restoration and spontaneous succession is compared (Figure 3). Although the time elapsed between the start of the successional process and active restoration is almost double, the fact is that both the value of *c* and the SAR are very close in the two types of plots. Despite these differences, the most important point is that the strategy followed in active recovery has been successful. The use of gypsum residues has not only maintained the originally planted species, but also, in this case, there are no invasive species alien to the gypsum ecosystems. However, some of the native ruderal species are also present in these plots, especially *Dittrichia viscosa*. As expected, the restored plots also have a higher value for *c* than the successional plots and their SAR curve is above it, although throughout the chronosequence it is observed how the regression lines of the successional plots tend to ascend (Figure 3).

The NMDS (Figure 4) shows a clear separation between the three types of plots studied, especially along a recovery gradient on axis 1. At the extremes of this axis are the undisturbed scrubland, and in the other, the initial plots of the successional chronequence. Those related to active restoration can be found between both groups of plots. The second axis establishes a gradient that is not as evident as the previous, but that has to do with the presence of species that can be considered ruderal, infrequent on gypsum, yet with a great invasive capacity, as is the case of *D. viscosa* or *P. miliaceum* (Figure 4). In addition, this second gradient seems to be related to the greater predominance of *G. struthium* subsp. *struthium*, and the presence of some annual species that, like *S. tenerrimus*, are rare in the scrubland plots. The PERMANOVA analysis also clearly supported the differentiation of the three types of environments studied (Table 3).

### 2.3. Succession Patterns in Other Gypsum Quarries Distributed throughout the Iberian Peninsula

As has already been pointed out and in accordance with the stated hypotheses, it was considered appropriate to ask whether the successional pattern so characteristic of the Almerian gypsum may be analogous to that of other Iberian territories. As can be seen in Figure 5, the general pattern observed throughout the Iberian geography is consistent with that described in this research for the successional plots of the Gypsum Karst Natural Park of Almería. In line with this pattern, *Gypsophila* species were predominant during spontaneous succession and much more abundant there than in the undisturbed scrub. Unlike the species of this genus, the rest of the gypsophytes clearly decreased in abundance. Another noteworthy aspect is that non-gypsophile or gypsovag species are often more abundant in gypsum scrub than gypsophytes themselves.

## 3. Discussion

### 3.1. Successional Plots

According to Scopus, 15–30 articles per year have been published on primary succession since 2000. However, very few of them were dedicated to quarrying [16]. In the case of gypsum quarries, from the first study by Mota et al. [24], who revealed spontaneous succession in this type of environment, the subject has been addressed by Mota et al. [20,45,47,48], Dana and Mota [21], Ballesteros et al. [49,50,51], Cañadas et al. [52], Foronda et al. [53], Pérez-García et al. [54] and Lorite et al. [55,56]. However, it should be noted that, although all the studies mentioned are of great value, none of them are based on the monitoring of permanent plots. Consequently, until now, the changes in the floristic richness, composition and diversity throughout the successional process in these post-mining environments have not been documented in detail. This is not a minor issue given that it is a priority habitat in the EU, the exploitation of which affects its endemic and threatened species.

The detailed monitoring of the spontaneous succession allowed identifying up to four typologies in the abundance patterns of the species recorded during the study period: 1—Primocolonizing species (pioneer) and those of early appearance during the successional process, abundant or at least frequent since the start of monitoring (e.g., *G. struthium* subsp. *struthium*, *S. gypsicola*, *D. glomerata*, *D. viscosa*). 2—Species of progressive colonization, present from the beginning, although not very abundant, with a clear tendency to increase over time (e.g., *L. fragilis*, *H. syriacum*). 3—Persisting species, with abundance and low frequencies throughout the period studied, although also present from the beginning (e.g., *H. squamatum*, *S. viscosa*). 4—Late species that were only recorded in the last years of monitoring (e.g., *H. alypoides*, *E. fragilis*). In any case, the nine perennial gypsophytes that are part of these gypsum scrubs, and are usually abundant in them, were present in the successional plots. However, the abundance of these species, with the exceptions of *G. struthium* subsp. *struthium* and *S. gypsicola*, was much lower than in the undisturbed scrub. This was the case for the two threatened local endemics *H. alypoides* and *T. turredanum*. The same happened with *S. viscosa* and *C. hispanica*, regional endemics, and with two of the most widely distributed gypsophytes in the Iberian Peninsula, *H. squamatum* and *O. tridentata* [45]. This last species, due to its size, N fixation capacity and its contribution of organic matter [57], can be considered of great interest in restoration. This disproportion in the abundance of the nine gypsophile species that was observed when comparing the successional plots with the scrub is amended, as is logical, in the restored area. Be that as it may, the set of gypsophyle species characteristic of these scrubs [45] appears in its entirety during spontaneous succession and the same occurs for the main accompanying and co-dominant species, such as in the cases of *S. tenacissima*, *H. syriacum*, *S. sediforme*, *L. fragilis*, *T. hyemalis* subsp. *hyemalis* or *A. terniflora*. The role of endemic species during succession, as has also been documented on islands [58], is a noteworthy aspect. Another no less trivial factor is that no species that could be considered invasive were found in the plots studied. This last feature, together with the previous two, the presence of gypsophytes and co-dominant species of the scrub, could be considered as indicators to evaluate ecological restoration success [59,60] since they meet the criteria established by Dale and Beyeler [61].

The richness and diversity indices of the successional plots experienced a progressive increase throughout the years of monitoring, although with slight fluctuations probably due to variations in rainfall that especially affected therophytes, as in the year 2014 (125 mm compared to 232 mm on average) (Appendix A). Regardless of these oscillations, the progressive increase in the number of species is also recognized in coefficient *c* of the SAR regression lines. This parameter, as already mentioned, is a surrogate of α-diversity [62] and showed its maximum values in almost all the plots at the end of the monitoring, years 2020 and 2021, with averages > 2. Although these values were much lower than those obtained for the gypsicolous scrub (Figure 3), they are close to those of the restored plots. These data point to the interest that passive restoration strategies may have in the case of open-pit gypsum mining.

### 3.2. Insights for Ecological Restoration

The progressive increase in floristic richness and diversity throughout spontaneous succession, as well as the floristic convergence towards the undisturbed scrubland, show the significant resilience (defined as the degree and speed at which an ecosystem recovers its initial structure and function after a disturbance, sensu Westman [63]) of the gypsum ecosystem studied. The rate of recovery of an ecosystem is affected by its intrinsic resilience, the level of human degradation and the features of the surrounding landscape [64]. These criteria must be considered in gypsum quarries since, as has been pointed out, auto-succession processes are spontaneously triggered there [24,48]. The decision about which restoration strategy to employ in a degraded system depends on its natural rate of recovery and the desired end point for the ecosystem.

The essential question is whether passive restoration (spontaneous succession or natural regeneration) can produce the desired goals efficiently, that is, within an acceptable period of time and at a lower cost relative to an active intervention. Much of the published work on restoration argues that the choice between passive restoration and technical recovery (active restoration) depends on both the intensity of the disturbance and the size of a disturbed site [60]. If the severity of the disturbance is low, there will be many surviving individuals among those that made up the natural vegetation, and recovery will be rapid. If the area is small, the limitations imposed by dispersal will also be fewer, even after severe disturbance [65]. Technical reclamation or active restoration [66] usually involves heavy interventions such as the restructuring of geomorphological features, the importation of soil and the planting or seeding of all plant species. According to these ideas, a priori, active restoration would be appropriate for the case at hand, given that the quarry in which the work was carried out covers a large area, almost 70 ha, and was intensely disturbed by human action [67]. Additionally, in this case, the target ecosystem might not be the original ecosystem, completely wiped out by mining activities. From this perspective, the so-called “reclamations” approach could be taken account of [68]. However, the type of restoration evaluated in this research can only be partially considered of this type, since in this case it did have as its main objective the restoration of the original ecosystem and its functionality. For this reason, the gypsum-based waste was mainly used as substrate, trying to emulate gypsic soils, and the plantation only included the original flora.

Reviewing the results shown in Appendix A, Figure 2 and Figure 3, the need for active intervention is not that evident, since spontaneous succession (closely related to the passive restoration approach) was relatively quick and successful. Not surprisingly, there is intense debate about which of these two opposing strategies should be employed [43], given the numerous examples of ecosystems recovering over a period of decades without intervention [69]. However, there are other considerations associated with passive restoration, not the least of which is the longer recovery time typically required for passive restoration, which could be perceived as a project failure, especially when compared to nearby active restoration efforts. Another point, no less important and also highlighted by Holl and Aide [43], is that passive restoration may be interpreted as an abandonment of the land. In the case of gypsum quarries, the visual impact is dramatic and can lead to a feeling of neglect. This is what seems to be shown by the fact that an unauthorized rave was held in the quarry studied during Christmas 2021 and that, previously, the area was also affected by the filming of the *Game of Thrones* series. Better signposting of the permanent plots and their monitoring could help to overcome this inconvenience. Another reason among those pointed out by Jones and Schmitz [69], is the fact that the study may not have been conducted over a long enough a time scale to detect recovery.

Between the two extremes indicated, spontaneous succession (passive restoration) and active restoration, intermediate interventions or what has been called assisted restoration [70] are also worth considering. In mining and industrial sites, assisted restoration aims to accelerate the natural regeneration of the ecosystem, which could otherwise be very slow, for example due to adverse conditions (very cold or dry sites, poor in nutrients, polluted). These types of actions may include an ecologically justified improvement of the abiotic conditions of the site, the suppression of undesirable species, the planting or seeding of target species and the creation of microsite heterogeneity. In the case of gypsum outcrops, the improvement of abiotic conditions must be interpreted with caution. In fact, if this improvement is understood as achieving a more fertile soil, rich in elements such as N and P, which are very scarce in gypsisols [71,72], interventions may lead to *ecce homo* (https://en.wikipedia.org/wiki/Ecce_Homo_(Mart%C3%ADnez_and_Gim%C3%A9nez), accessed on 30 December 2022). This type of restoration can generate communities completely alien to gypsum habitats, in which halophilic species such as *Atriplex halimus* become predominant [73]. Perhaps the most accurate term to describe the restoration performed here is assisted restoration. Since this intervention occupies almost 20 ha, it can be said that it is the most extensive ever known, and perhaps one of the few documented examples that exist on a global scale on these ecosystems. Furthermore, due to its characteristics, this restoration was moderate, since no actions were carried out such as substrate fertilization or the elimination of competing plants by chemical or physical weeding, and support irrigation was only implemented during planting and the first summer. Due to these peculiarities, only some species outside the undisturbed scrubs, such as *D. viscosa*, proliferated moderately after the intervention developed.

Regarding the range of the *z*-values obtained, it was very narrow for the successional plots, between 0.36 and 0.39 with a mean value of 0.37. In the case of restoration, this last value was 0.41 and 0.23 for the scrub. This is consistent with the general range of 0.2–0.4, commonly observed for the *z*-values of SARs modelled by a power function. An important aspect is that a slight decrease in *z* was detected as primary succession progressed over time, more markedly when only perennial species were considered. This was probably due to the fact that the presence and abundance of perennial species was less conditioned by the amount of rainfall (Appendix A). This pattern is very similar to that found by Anthelme et al. [74] and earlier by Osbornová et al. [75], Lepš [76], and Lepš and Štursa [77]. However, these last three papers do not deal with primary succession, and in the first one no interpretation of *z* is made in relation to the progression of the sequence. Most of the research that has focused on the *z*-value has also studied SAR and island biogeography. In the first case, in relation to the rate at which species are added as the area increases and, in the second, to differentiate various types of islands, from true islands to patches of habitats [78]. In relation to the latter, interpreting an area completely devoid of vegetation and subjected to extreme environmental conditions (without soil or nutrients, with extreme nutritional imbalances, without the capacity to retain water…) as if it were an island habitat is tempting. The parallel between the arrival of propagules on an ecological island and spontaneous succession seems appropriate. According to Fattorini [79], higher immigration rates should increase *z*-values. Following this same reasoning, as time passes, the exact opposite would happen with the successional process, as Cramer and Hobbs [80] point out in relation to the founder-to-dominance-controlled communities model proposed by Yodzis [81] which predicts the decrease in the *z*-value of the species’ area curve. The aforementioned articles and the results obtained in this study lead to this conclusion. In any case, and awaiting future research that will deal with this aspect in more detail, the authors consider that the results obtained here are of great interest and can lay the foundations for new experimental designs to obtain more conclusive evidence.

However, one last comparison is missing to answer the question of whether restoration, passive or assisted, achieves the objective of recovering biodiversity following mining. Figure 3 shows the results for species’ richness and the Shannon index in the studied plots. From the comparison between the values obtained for the reference scrubs and those of the other two environments, it can be deduced that the former are higher. However, it is evident that during spontaneous succession, the values rise progressively. This increase is linear over the time monitored, with a strong fit in the case of perennial species (Appendix A), suggesting that it could progress to levels close to those of undisturbed scrub. If these linear relationships are assumed over time, and considering the values obtained for *z* and *c*, the recovery process of the final pit of the quarry could take more than 150 years.

### 3.3. Succession Patterns in Gypsum Quarries in the Iberian Peninsula

Since the pattern found for spontaneous succession, at least when a species of the genus *Gypsophila* is present, is generalized for all the outcrops studied which were exploited by mining, it seems evident that it can be used to propose restoration strategies throughout the Spanish territory. This consideration is of great relevance even in a European context and, especially, in relation to the Habitats Directive within which it is essential to develop robust restoration plans [82,83]. It is worth emphasizing here that different species of this genus are widely distributed in territories with extensive gypsum outcrops, such as Italy (especially in Sicily), Iran or Turkey. In this last country, spontaneous succession processes have been observed similar to the one described here, carried out by *G. eriocalix* Boiss.

## 4. Materials and Methods

### 4.1. Study Area and Data Collection

#### 4.1.1. Location of Permanent Plots for Diachronic Study

The study area is located in the Natural Protected Area of the Gypsum Karst Natural Park in Sorbas (Almería) and particularly in the Majadas Viejas mining lease (Figure 6). This is also a Special Protection Area (SPA), under the European Union Directive on the Conservation of Wild Birds, and Special Area of Conservation (SAC) within the EU’s Natura 2000 Network (Cod. ES6110002). A detailed description of the environmental conditions of this site can be found in Mota et al. [45] and Merlo et al. [84].

The quarry where the study was conducted affects two mining grids (each between 27 and 28 ha), in the Peñón Díaz and Cerrón del Huelí areas, where approximately 50 ha have been altered since the beginning of the mining process. In 2009, ten permanent plots were established in a flat area where exploitation had ceased eight years earlier (Figure 6). The sample plots have dimensions of 20 × 50 m (1000 m^2^) and include nested subplots of 1 m^2^ (*n* = 10), 10 m^2^ (*n* = 2) and 100 m^2^ (*n* = 1) [85]. In each of these subplots and in the total plot (Figure 6), the presence of the different species of vascular plants, both perennial and annual, as well as their cover (%), was recorded from 2009 to 2021. Detailed information on the flora and vegetation of this area can be found in Mota et al. [45]. The botanical nomenclature for the recorded species follows that of Flora Iberica [86].

In addition to these plots, another ten plots were established following the same design, five within the same mined area and five in its vicinity. Five of them corresponded to an actively restored area within the quarry. In this case, native species obtained in a nursery from seeds collected in situ were used. Before this planting process, the pit floor was covered with topsoil (soil from the ridges of the exploitation fronts) and later with “fines”, waste resulting from the crushing of the gypsum, and rich in this material, in which gravel-sized fragments predominate. To these actions, a slight remodeling of the land was added, almost flat in the areas where the sampling plots were placed, as well as terracing. After planting, the plants received support irrigation until the end of the first summer, since the characteristic dryness of this season for this territory implies a critical phase for them. In any case, the substrate was neither enriched with fertilizers nor was there weeding of any kind. These operations were carried out in 2011. The other five established plots belong to the natural (undisturbed) gypsicolous scrubland corresponding to the priority habitat of the EU Habitats Directive number 1520*. The floristic composition of the plots studied is detailed in Appendix A.

#### 4.1.2. Study of the Successional Pattern in Iberian Gypsum Quarries

Previous research already suggested the great resilience of priority habitat 1520* [21,47]. In order to verify whether the successional pattern found was geographically recurrent, and not only restricted to the one found in the Karst Natural Park in Yesos de Sorbas, the study area was extended to the rest of the Iberian territories with gypsum areas (Appendix A). The answer to this question is not obvious, given that since the pioneering observations of Cavanilles at the end of the 18th century, gypsicolous scrublands are known to exhibit a high ß-diversity due to the presence of numerous regional and local endemics [45]. This set of floristic combinations provides a complex puzzle for the self-assembly of gypsum communities after disturbances and especially in relation to their resilience. Due to the extension of the geographical area studied, to sample the vegetation, plots of 5 × 5 m were used here, in which both the presence and the coverage of the species present (%) were recorded. In this case, there were two types of plots: those located in gypsum quarries where spontaneous succession was observed and those in the surrounding unaltered gypsum scrub. In these plots, as in the case of the successional ones, the samples were taken at the end of May and June, when all the species, including the annual ones, can be identified. In total, 28 samples were taken from quarries, taking advantage of all the locatable sites, and 78 from the surrounding gypsicola scrub (Table 4).

### 4.2. Data Analysis

The diversity associated with the sampled plots was calculated using the species-area relationships or SARs [87]:*S* = *c* · *A^z^*,(1)
where *S* is the number of species, *A* is the area, *c* is a coefficient and *z* is the slope of the line. In this model, the intercept (*c*) measures α-diversity (richness), and the slope (*z*) measures β-diversity (a measure of the differentiation between habitats or habitat fragments) [78]. This latter parameter also represents an estimate on the degree of insularity of fragmented or disjointed territories, as is the case of gypsum outcrops [88]. This characteristic of gypsum habitats [45,89,90,91], its good fit to the type of sampling used and the possibility of making both graphic and numerical comparisons among plots, justify the use of Arrhenius’ function. In addition, this model allows representing the curves and graphically displaying the temporal trajectory of the successional plots, according to their floristic composition, in relation to the reference scrubland.

To calculate and compare the richness and diversity of the plots in the three environments studied, in addition to the *c* parameter, species richness and the Shannon and Simpson indices included in the statistical analysis packages were used: Statgraphics, PCord [92], PAST [93] and Primer-e [94].

Regarding the floristic composition of the plots, and the abundance of the species, different multivariate analyses were carried out, such as NMDS (Nonmetric multidimensional scaling), and PERMANOVA, using the aforementioned software and pre-transforming the data (square-root) to smooth out the different orders of magnitude in species’ abundance and the frequent absence of many species from the inventories. NMDS plots provide a powerful representation of sample patterns, and its interpretation is straightforward. In this study’s analysis, the closer the samples are to each other, the more similar their floristic composition is. To refine the NMDS results, PERMANOVA was used here to contrast plots that belong to groups inherent to the study design, such as different years after disturbance and emplacement.

Finally, to examine whether the successional pattern found in the quarries in the area studied was repeated on an Iberian scale, the species found were grouped into non-gypsophiles and gypsophiles, according to Mota et al. [45] and Musarella et al. [95], although within the latter the species of the genus *Gypsophila* were separated into their own group since it is a gypsophyte with a great colonization capacity [20,47,48].

## 5. Conclusions

This research documents for the first time and through the use of permanent plots (13 years of monitoring) the direct primary spontaneous succession process in gypsum quarries after the cessation of mining exploitation and has revealed the great resilience of these ecosystems. In general terms, this process stands out for the presence of almost all the gypsophile species of the area and, very especially, for that of *Gypsophila struthium* which, unlike the rest, is not only present from the first phases of the succession, but whose abundance is much higher in post-quarrying areas provided the substrate is gypsiferous. These characteristics reveal two very original features of the primary succession in gypsum that have to do with the insular character of the outcrops (geological island) of these ecosystems and the limited pool of species capable (dominated by endemic gypsophiles) of establishing themselves ex novo in such an extreme environment.

Since the gypsum habitat is a priority for the EU (Habitats Directive), spontaneous succession can be of great interest for post-mining restoration. This research has proven that a practical approach to primary autosuccession may be as interesting as active restoration, given that the rate of ecosystem recovery is comparatively fast and effective. According to the results obtained, the trajectory of primary succession leads towards the reference gypsicolous scrub. This is shown by the clearly recognizable floristic changes in the successional plots for intervals of 4 or 5 years. This conclusion contradicts what could be expected a priori due to the enormous intensity of the disturbances that quarrying entails, which not only affects the biotic component but also the landscape and its geomorphology. Due to the uniqueness of these ecosystems, an assisted restoration strategy that combines autosuccession with the use of gypsum waste can also help manage mining waste not only to maintain the genuine floristic composition of priority gypsicolous scrub, but also for landscape remodelling. The management of these mining remnants is key and must take into account the role of gypsum in substrate preparation, otherwise the risk of distorting the original ecosystem is obvious.

SAR has been shown to be very useful for detecting the functioning of the gypsum ecosystem, especially through parameters *c* and *z*, related to α-diversity and ß-diversity. Not only do both parameters serve to describe the characteristics and trajectory of spontaneous succession, but they are also useful for evaluating the success of restoration actions.

The successional pattern recorded in the Sorbas Gypsum Karst Natural Park is repeated throughout the Iberian geography wherever a species of the genus *Gypsophila* is present, which leads to the conclusion that the validity of passive restoration is applicable to other parts of Spain where there are open-cast gypsum operations. For this, it would always be necessary to leave patches of surrounding vegetation where the reference ecosystem is well represented, which can serve as models and a source of propagules, especially seeds, for restorative actions.

## Figures and Tables

**Figure 1 plants-12-01162-f001:**
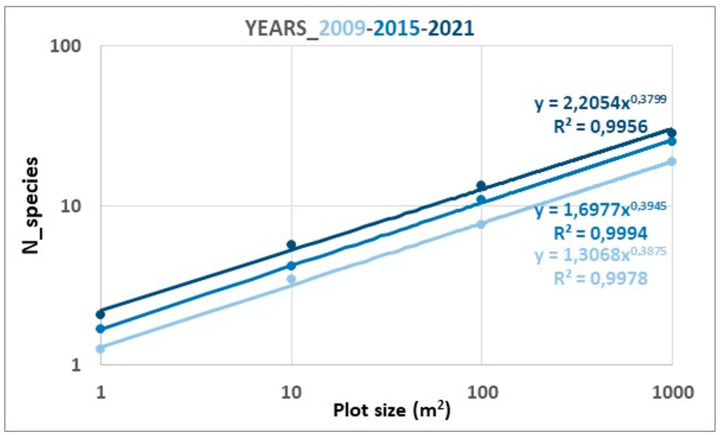
SAR curves for the ten successional plots during the years 2009 (initial), 2015 (mid-term) and 2021 (last).

**Figure 2 plants-12-01162-f002:**
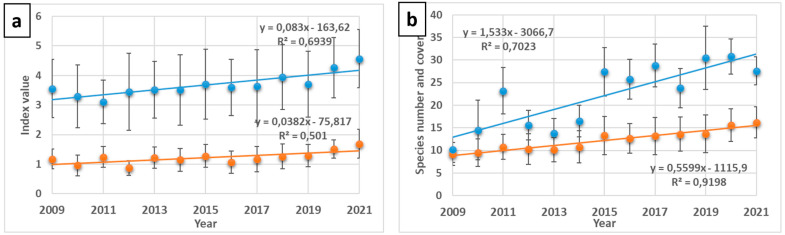
(**a**) Variation in the mean number of species (blue line) and their coverage (red line) during the 13 years of monitoring; (**b**) The same for the Simpson (red line) and Shannon (blue line) indices. Bars correspond to the standard deviation.

**Figure 3 plants-12-01162-f003:**
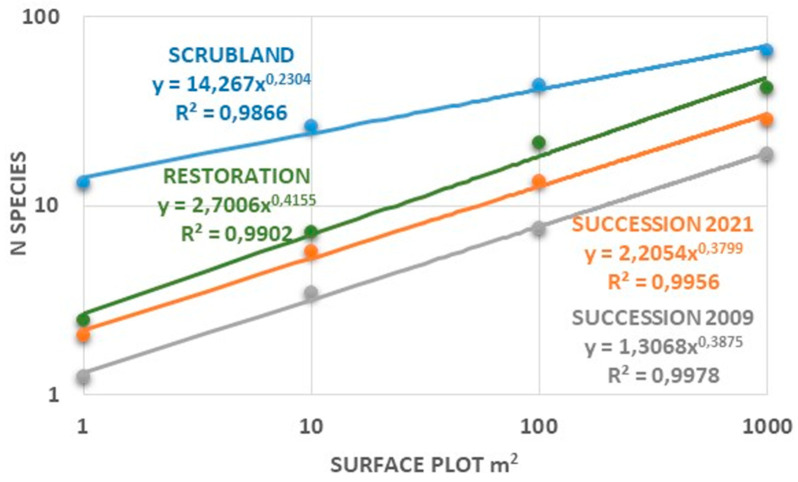
SAR Comparison between the three regression lines obtained for the successional plots (year 2021), active restoration (or assisted) and the reference scrubland.

**Figure 4 plants-12-01162-f004:**
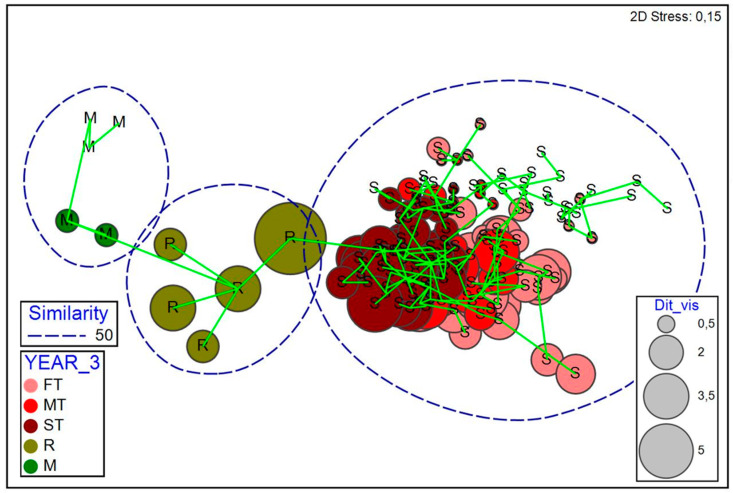
Non-metric MDS for the studied plots. M = scrub plots; R = restoration plots; S = plots of succession. Pink indicates the successional plots from 2009 to 2013 (FT); red from 2015–2017 (medium-term plots, MD); dark red from 2017 to 2021 (ST). The green colours correspond to the scrub and restoration plots. The size of the circles expresses the abundance (canopy cover percentage) of the ruderal species *Dittrichia viscosa*. The blue dashed line shows the samples with a similarity level > 50%. The minimum spanning tree superimposed on the image connects the different sampled plots.

**Figure 5 plants-12-01162-f005:**
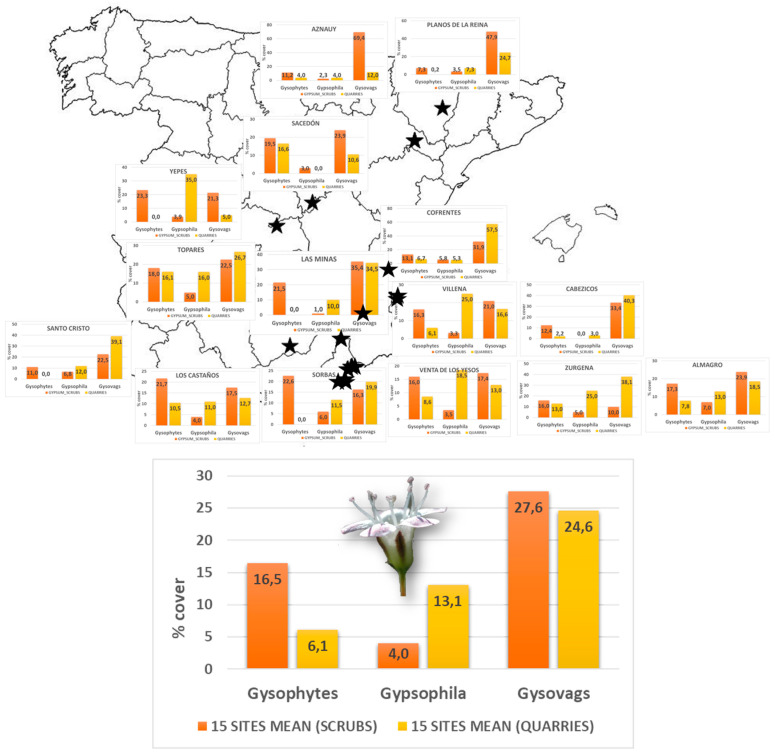
Mean values of the cover of the gypsophile species or gypsophytes (except the species of the genus *Gypsophila*), *Gypsophila* spp. and non-gypsophile species (gypsovags) in undisturbed scrub (orange) and successional scrub in quarries (yellow) in the Iberian gypsum areas.

**Figure 6 plants-12-01162-f006:**
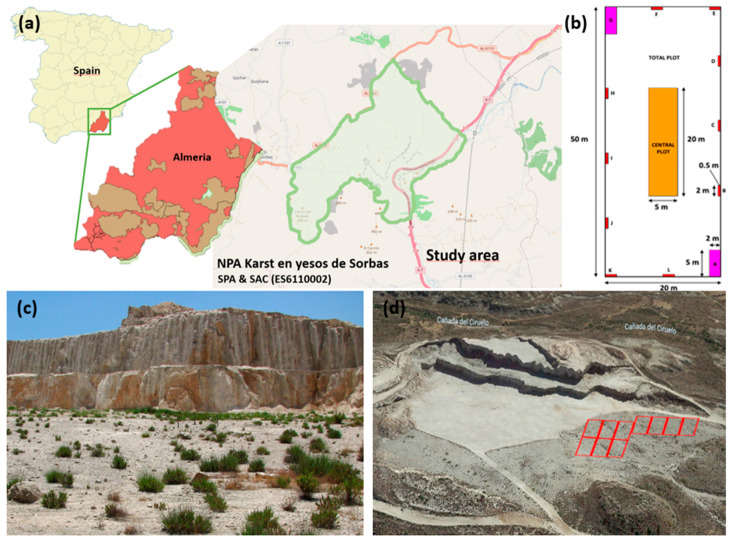
(**a**) Location of the area studied, next to the gypsum karst of Sorbas, in the province of Almería; other protected areas are in brown; (**b**) Schematic of a sample plot with nested subplots; (**c**) Quarry bottom appearance during spontaneous succession; scrubland dominated by *Gypsophila struthium* subsp. *struthium*; (**d**) Panoramic view of the quarry with the old exploitation front and the area where the permanent plots were located (https://www.google.com/intl/es/earth/, accessed on 30 December 2022); aerial views of the restored areas can also be seen with the plants regularly arranged.

**Table 1 plants-12-01162-t001:** PERMANOVA for years (YE).

Source	df	SS	MS	Pseudo-F	*p* (perm)	Unique Perms
YE	12	31,670	2639.2	31.295	0.0001	9787
Res	117	98,668	843.32			
Total	129	130,000				

**Table 2 plants-12-01162-t002:** PERMANOVA for plot order (OR).

Source	df	SS	MS	Pseudo-F	*p* (perm)	Unique Perms
OR	9	65,641	7293.4	13.528	0.0001	9839
Res	120	64,698	539.15			
Total	129	130,000				

**Table 3 plants-12-01162-t003:** PERMANOVA analysis for the three types of environments considered (spontaneous Succession-S, restoration-R and undisturbed scrub-M).

							PAIR-WISE TESTS			
Source	df	SS	MS	Pseudo-F	*p* (perm)	Unique Perms	Groups	*t*	*p* (perm)	Unique Perms
SI	2	214.25	107.13	22.173	0.0001	9894	S, R	3.5393	0.0001	9916
Res	137	661.89	4.8313				S, M	5.8556	0.0001	9907
Total	139	876.14					R, M	2.2991	0.0093	126

**Table 4 plants-12-01162-t004:** Plots in Spain of the gypsum quarries and those in the surrounding unaltered gypsum scrub.

Province	Locality	Quarries	Scrubs
Albacete	Las Minas	1	5
Alicante	Cabecicos de Villena	3	6
Alicante	Villena	1	4
Almería	Los Castaños	4	4
Almería	Sierra de Almagro	3	3
Almería	Sorbas	2	2
Almería	Topares	1	2
Almería	Venta de los Yesos	2	2
Almería	Zurgena	1	1
Guadalajara	Sacedón	1	9
Huesca	Azanuy	1	9
Huesca	Planos de Elena	3	6
Jaén	Cabra de Santo Cristo	1	8
Toledo	Yepes	1	13
Yepes	Cofrentes	3	4

## Data Availability

Original data is included in Appendix A and any other data that is required will be provided.

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
