# Peer review of "Spontaneous Primary Succession and Vascular Plant Recovery in the Iberian Gypsum Quarries: Insights for Ecological Restoration in an EU Priority Habitat"

_plants, 2023, doi:10.3390/plants12051162_

Round 1

Reviewer 1 Report

The manuscript presents important results that will guide restoration practices of gypsum quarries. I found the study to be of good quality. Especially since the findings are deduced from long-term monitoring.

Please see the annotated PDF with suggestions and corrections.

Author Response

Reviewer_1

Abstract

Line 10.- Abstract changes have been accepted.

Introduction

Line 38.- We maintain the definition offered on ecological succession, which does not differ much from the one offered by reviewer_1, and emphasises an important idea in the manuscript: disturbance.

Lines 63-64.- The sentence pointed out by the reviewer has been deleted and the associated references have been eliminated.

Line 64.- To clarify the sentence, the word "successional" has been added to "pattern".

Line 102.- According to Svoboda and Henry, in extreme habitats only a group of species is available to participate in succession processes. This is a key aspect and pattern in gypsum outcrops.

Line 105.- Based on the reviewer's suggestion, we have shortened the sentence so that it is not repetitive.

Line 108.- Fixed.

Line 114.- We have rewritten and improved the paragraph.

Line 118.- Deleted.

Line 122.- Yes, the number is necessary since it is the one that appears in the EU Habitats Directive. We have tried to make this aspect clear.

Line 124.- "this research" has been changed to "the study".

Lines 138-139.- We have introduced small changes in the formulation of the objectives that will help to better understand point 4.

Results

Line 142.- Indeed, as the reviewer suspected, the Material & Methods section is located in another part of the article in accordance with the journal's guidelines. However, we understand that a previous mention of the SAR curves, before stating the objectives, may help clarify some relevant aspects of our manuscript without the need to previously read the material and methods section. For this reason, we have included a brief sentence between lines 118-119.

The study hereby presented provides detailed information on spontaneous primary autosuccession, after monitoring ten permanent post-mining plots for 13 years. For this, SARs (species-area relationships) curves and diversity indices were used.

Line 145.- Deleted.

Line 148.- The scientific names of the species have been arranged alphabetically.

Line 151.- Deleted.

Line 168.- It means that they have a completely different behavior, but we have rephrased it.

Line 173.- We have changed "along time" to "with time". However, in the following sentence, "present" is not the same as "ubiquitous". It is not just that Diplotaxis harra is present, but that it is present in each and every plot.

Line 179.- Deleted.

Line 216-217.- We have added the error bars and corrected “covert” for “cover” in the graph legend. We have added the standard deviations and we have clarified that it is the mean number of species:

Figure 2. (a) Variation in the mean number of species (blue line) and their coverage (red line) during the 13 years of monitoring; (b) The same for the Simpson (red line) and Shannon (blue line) indices.

Lines 232 and 235.- We have added “active”.

Line 250.- Fixed, now it says “along a recovery gradient on axis 1”.

Line 279.- Done.

Line 281 (figure 5).- Fixed: cover and gypsovags.

Lines 286-287.- Sentence “Studies of primary succession and plants include classic papers on dunes [4,50,51], volcanoes [52,53,54], and glacial moraines [55,56]” has been removed and the following sentence has been rephrased “According to Scopus, between 15-30 articles per year have been published on primary succession 2000”.

Lines 289-294.- We fully understand the meaning of the suggestion made by the reviewer. We totally agree that this paragraph could be removed from the manuscript without hardly altering it at all. However, we understand that it is a recognition of the merits of other colleagues who, although in a very different way, have been concerned with the succession or restoration of gypsum quarries. For this reason, we consider that it is best to keep this paragraph in the final version of the manuscript, since these papers are only mentioned at this point. All these authors are, moreover, potential readers of this manuscript. In addition, they will be aware of its existence precisely because they are cited in it.

Line 297.- We have replaced “whose” with “the exploitation of which”.

Line 299.- We have replaced "allowed to recognize up to" with "identified".

Line 309.- Done.

Line 312-313.- Indeed, the use of “endemics” is more common than “endemism”. Fixed.

Line 318.- If the word "corrected" is replaced by "amended" in the manuscript, everything becomes clearer. Fixed.

Line 319.- Done.

Line 341.- Indeed, it is a very interesting paper that we have included among the references.

Line 343.- We have changed "great" to "significant”.

Lines 364-365.- We fully agree with this reviewer's comment. Perhaps our insistence on floral composition may lead one to think that we do not consider functionality important, but that is not the case. We simply consider that composition and functionality are linked. This idea is reflected in the manuscript, but we have tried to express it more clearly. For this, we have added this idea in line 368 (...the restoration of the original ecosystem and its functionality).

Line 387.- Done.

Line 458.- In this case, the name of the author is included because unlike the other taxa mentioned in the text, this species is not listed in Appendix I (S1). It is a species from the eastern Mediterranean.

Line 459.- That is correct.

Line 462.- We have replaced "The area studied" with "Study area".

Lines 470-471.- Fixed.

Lines 479-480.- We have clarified that it is "Five of them corresponded to an actively restored area within the quarry".

Line 482.- Deleted.

Line 491.- This point has already been clarified in the comment on line 122.

Line 507.- Fixed and we have replaced “endemisms” with “endemics”.

Line 515.- Done.

Lines 521-522.- We have rewritten the paragraph and eliminated some parts. Finally it is as follows: The diversity associated with the sampled plots was calculated using the species-area relationships or SARs.

Line 527.- This refers to beta diversity. We think this is clear.

Line 560.- Done.

In addition to all these changes, the entire text of the manuscript has been thoroughly revised again.

Reviewer 2 Report

It is a well-designed and written paper that deals with spontaneous succession in gypsum quarries after the cessation of mining exploitation. The results are based on long-term studies conducted on permanent study plots, and it is worth underlining that the Authors compared communities that formed during succession with plots associated with active restoration and with predominant natural (reference) vegetation. The result of this paper can be beneficial in planning reclamation /restoration of post-industrial sites.

Below I included some minor suggestions.

Keywords:

Line 24 why do you use * with gypsophil

Line 67 use creates a new site instead of new surface

Line 90 after four check space

I prefer the layout where Materials and Methods are after Introduction and before results but I know that it is layout of the manuscript

Figure 5 the figures (bars)  in the upper part need some correction since it is difficult to read the values on them 

Line 143 I suggest changing the title into The successional patterns observed on permanent study plots in gypsum quarries in the Iberian Gypsum quarries: Insights for Ecologcial Restoration in an EU prority Habitat.

Line 477 maybe it is better to  use the word recorded instead of registered

Line 510 how the coverage of species was determined (in percentage, in Braun-Blanquet scale

Table 4 According to caption column Quarries should be before Scrub

Line 536 Be precise what richness and rarity indices were used

Line 541 what kind of transformation was used

Line 360 what Authors mean using geographical features land relief ? 

Importation of soils – from what soils are imported, mainly if the area of quarry is large. We sometimes talk about technical versus biological soil reclamation

Line 399 check spacing 

Author Response

Reviewer_2

Line 24.- (Keywords) Why do you use * with gypsophil? We found it to be an ingenious solution that encompasses terms such as "gypsophile", "gypsophily", "gypsophilic", "gypsophilous", ... And it works in bibliographic searches! It will be easier for other colleagues to find our manuscript.

Line 67.- Use creates a new site instead of new surface. Done.

Line 90.- After four check space. Fixed.

I prefer the layout where Materials and Methods are after Introduction and before results but Iknow that it is layout of the manuscript. We do too, but we follow the guidelines of the journal.

Figure 5 the figures (bars) in the upper part need some correction since it is difficult to read the values on them. We absolutely agree. Thank you for this important observation. We have reworked the figure.

Line 143.- I suggest changing the title into The successional patterns observed on permanent study plots in gypsum quarries in the Iberian Gypsum quarries: Insights for Ecological Restoration in an EU prority Habitat. We appreciate the comment, but we prefer to keep our title.

Line 477.- Maybe it is better to use the word recorded instead of registered. Done.

Line 510.- How the coverage of species was determined (in percentage, in Braun-Blanquet scale. It is a visual estimate of percentage.

Table 4.- According to caption column Quarries should be before Scrub. Fixed.

Line 536.- Be precise what richness and rarity indices were used. We have used the number of species, the Simpson index and the Shannon index. We have also reflected this in the text of the manuscript.

Line 541.- What kind of transformation was used? Square-root transformation. We have also written it like this in the text of the manuscript.

Line 360.- What Authors mean using geographical features land relief? Actually, we meant to say "geomorphological". Fixed.

Importation of soils – from what soils are imported, mainly if the area of quarry is large. We sometimes talk about technical versus biological soil reclamation. This aspect is explained in lines 364-365: “For this reason, the gypsum-based waste was mainly used as substrate, trying to emulate gypsic soils, and the plantation only included the original flora”.

Line 399.- Check spacing. Done.

In addition to all these changes, the entire text of the manuscript has been thoroughly revised again.